# Novel Bioactive Adhesive Monomer CMET Promotes Odontogenic Differentiation and Dentin Regeneration

**DOI:** 10.3390/ijms222312728

**Published:** 2021-11-25

**Authors:** Youjing Qiu, Takashi Saito

**Affiliations:** Division of Clinical Cariology and Endodontology, Department of Oral Rehabilitation, School of Dentistry, Health Sciences University of Hokkaido, Kanazawa 061-0293, Japan; t-saito@hoku-iryo-u.ac.jp

**Keywords:** adhesive monomer, biomaterials, dental pulp stem cells, dentin regeneration, odontoblasts

## Abstract

This study aimed to evaluate the in vitro effect of the novel bioactive adhesive monomer CMET, a calcium salt of 4-methacryloxyethyl trimellitate acid (4-MET), on human dental pulp stem cells (hDPSCs) and its capacity to induce tertiary dentin formation in a rat pulp injury model. Aqueous solutions of four tested materials [4-MET, CMET, Ca(OH)_2_, and mineral trioxide aggregate (MTA)] were added to the culture medium upon confluence, and solvent (dH_2_O) was used as a control. Cell proliferation was assessed using the Cell Counting Kit-8 assay, and cell differentiation was evaluated by real-time quantitative reverse transcription-polymerase chain reaction. The mineralization-inducing capacity was evaluated using alizarin red S staining and an alkaline phosphatase activity assay. For an in vivo experiment, a mechanical pulp exposure model was prepared on Wistar rats; damaged pulp was capped with Ca(OH)_2_ or CMET. Cavities were sealed with composite resin, and specimens were assessed after 14 and 28 days. The in vitro results showed that CMET exhibited the lowest cytotoxicity and highest odontogenic differentiation capacity among all tested materials. The favorable outcome on cell mineralization after treatment with CMET involved p38 and c-Jun N-terminal kinases signaling. The nuclear factor kappa B pathway was involved in the CMET-induced mRNA expression of odontogenic markers. Similar to Ca(OH)_2_, CMET produced a continuous hard tissue bridge at the pulp exposure site, but treatment with only CMET produced a regular dentinal tubule pattern. The findings suggest that (1) the evaluated novel bioactive adhesive monomer provides favorable biocompatibility and odontogenic induction capacity and that (2) CMET might be a very promising adjunctive for pulp-capping materials.

## 1. Introduction

In clinical endodontics and pediatric dentistry, the surface between a material and oral tissue may be one of the most important interfaces. Living cells and tissues directly exposed to applied materials construct a complicated interacting network to regulate apoptosis and regeneration. With the development of dental science, material biocompatibility has been increasingly important for endodontic treatments because it is the key deciding factor for the success of vital pulp therapy (VPT), especially pulp capping [1]. Calcium-containing materials, such as various forms of calcium hydroxide [Ca(OH)_2_] and mineral trioxide aggregate (MTA), have been extensively used in VPT, and Ca(OH)_2_ has long been considered the “gold standard” [2]. However, the poor quality of the resulting dentin bridge and its lack of an inherent ability to adhere to dentin or other restorative materials are at the center of potential failure with this material because of its caustic effect and high pH [3,4]. MTA, which exhibits better biocompatibility and mechanical properties and induces a more favorable result than Ca(OH)_2_ cements, was introduced to the market as a Ca(OH)_2_ alternative [5,6]. Nevertheless, its drawbacks have urged the development of further ideal pulp-capping materials [7]. For example, compared to MTA, Biodentine (Septodont, Saint-Maur-des-Fossés Cedex, France) sets faster, exhibits less discoloration, and shows similar clinical results; thus, Biodentine is considered a suitable alternative to MTA [8]. And several experimental materials based on calcium-silicates and calcium-phosphates were reported previously [9].

Resinous materials have been widely used for endodontic treatments in recent decades and have been proposed as potential pulp-capping agents [10]. They exhibit several advantages, including their high mechanical strength, easy application, and ability to be light cured immediately, providing superior adhesion to peripheral hard tissues and an effective seal against microleakage. However, due to their potential toxicity, these materials represent a risk to pulp vitality [11]. The detrimental effect of resinous materials was suggested to primarily be a result of the diffusion of residual monomers from the material-tissue surface [12,13,14]. A previous study revealed that most of the monomers used in the adhesive system and composite resin, such as 2,2-bis [4-(3-methacryloxy-2-hydroxypropoxy) phenyl]propane (bis-GMA), triethylene glycol dimethacrylate (TEGDMA), and 2-hydroxyethyl methacrylate (HEMA), showed cytotoxic effects on odontoblast-like cells in vitro [15]. And cytotoxic effects can depend not only on the materials but also on the conversion within the environment [16].

CMET, a calcium salt of 4-methacryloxyethyl trimellitic acid (4-MET), was used as an adjunctive component in a commercialized adhesive system in 2019. In this novel monomer, the strong acidity of 4-MET is neutralized by the replacement of the hydrogen ions on the two carboxyl groups. CMET induces dentin remineralization in vitro and increases the shear bond strength, bending strength, and compressive strength of resin-based coating materials [17]. Long-term evaluation of eluate from CMET-adhesive hardening composite revealed remineralization potential and favorable micro-tensile bonding strength; moreover, the nanospace underneath the hybrid layer was blocked by the remineralization substance, revealing its potential as a dentinal hypersensitivity treatment [18]. CMET was also found to inhibit the formation of *Streptococcus mutans* biofilm, indicating its possible application in caries prevention [19]. Additionally, the biocompatibility of CMET on terminally differentiated postmitotic odontoblast-like cells was more favorable than that of Ca(OH)_2_ and MTA, as they are at the front line of the interface between material and dentin-pulp complex. CMET exhibits low cytotoxicity, high mineralization, and a high differentiation-inducing ability, which promotes intracellular Ca^2+^ homeostasis and provides a signal for the activation of downstream events, and is thus considered a suitable material for regenerative endodontic procedures [20]. In contrast to odontoblast-like cells, whose main function is to secrete reparative dentin under suitable stimulations, dental pulp stem cells (DPSCs) are capable of self-renewal and multilineage differentiation [21], and the ability of materials to aid or induce the healing process through odontoblast-like cell differentiation and maturation affects their bioactivity and biocompatibility. Under favorable conditions, DPSCs differentiate into odontoblast-like cells that produce regenerative dentin, establishing a new mineralized barrier, shielding the pulp from additional injury, and permitting healing and repair [22]. As the pulpal response toward CMET has not been examined in vitro or in vivo, whether this novel monomer increases the regeneration ability of the pulp remains unknown. Therefore, the objective of the present study was to assess (1) the in vitro effect of this novel monomer on human dental pulp stem cells (hDPSCs) compared with those of 4-MET, Ca(OH)_2_, and MTA and (2) its capacity to induce hard tissue formation in a rat pulp mechanical exposure model compared with that of Ca(OH)_2_.

## 2. Results

### 2.1. Cell Viability

As shown in Figure 1, low concentrations (below 20%, *v*/*v*) of CMET and MTA promoted cell viability. Ca(OH)_2_ at even a low percentage (5%, *v*/*v*) inhibited cell viability. The CMET group exhibited the highest cell viability and lowest cytotoxicity among the four groups.

### 2.2. Alizarin Red S (ARS) Staining

Except for 4-MET, which inhibited cell mineralization, the three other materials significantly accelerated calcium nodule formation of hDPSCs. Moreover, the CMET group exhibited a uniform mineralized appearance, but the mineral nodules formed in the Ca(OH)_2_ and MTA groups were similar to those upon the addition of external calcium ions such as a calcium chloride solution (Figure 2A). Inhibition of the nuclear factor kappa B (NF-κB) pathway reduced CMET-enhanced calcific deposition and inhibition of the p38 and c-Jun N-terminal kinases (JNK) mitogen-activated protein kinases (MAPK) pathways completely depressed hDPSCs mineralization.

### 2.3. Relative Alkaline Phosphatase (ALPase) Activity

ALPase activity assay indicated that CMET significantly increased hDPSCs ALPase activity on days 14 (Figure 2C) and 21 (Figure 2D) and subsequently decreased ALPase activity on day 28 (Figure 2E). MTA treatment slightly increased ALPase activity on day 21. The Ca(OH)_2_ did not positively regulate ALPase activity. Similar to the mineralization results, inhibition of the p38 and JNK MAPK pathways, especially the p38 pathway, depressed the CMET-induced increase in ALPase activity.

### 2.4. Real-Time Quantitative Reverse Transcription-Polymerase Chain Reaction (Real-Time RT-PCR)

CMET significantly enhanced the mRNA expression of all tested odontogenic differentiation markers, particularly the dentin-specific genes DSPP and DMP1 (Figure 3). The addition of Ca(OH)_2_ or MTA did not have such an effect. The ability of CMET to induce odontogenic differentiation was suppressed by the addition of an NF-κB pathway inhibitor.

### 2.5. Histomorphological Features

At 14 days after operation, severe inflammation was observed in the negative control group, as indicated by hyperemia and a highly disrupted morphology. The cells were flattened and had lost the typical columnar shape of odontoblasts, and irregular hard tissue formation was observed in the pulp horn area (Figure 4C). A dentin-associated partial hard tissue bridge was found near the exposure site in the Ca(OH)_2_-treated group, with a local inflammatory process (Figure 4D). The CMET group exhibited the presence of a thin complete bridge invading the pulp space to the opposite dentin wall, and the adjacent pulp tissue appeared normal (Figure 4E). On day 28, heterogeneous hard tissue beneath the secondary dentin was observed in the negative control group, excluding the exposure site. The columnar shape of the odontoblasts was still absent (Figure 4G). An irregular hard tissue bridge without an obvious tubular structure was observed in the Ca(OH)_2_ group (Figure 4H). A continuous thick dentin bridge and tertiary dentin were seen in the CMET group, with well distinguishable dentinal tubules. An intact odontoblastic layer aligned along the periphery of the pulp and adjacent to the dentinal bridge was observed (Figure 4I). Hard tissue bridges formed faster, were thicker, and had a better structure in the CMET group than in the Ca(OH)_2_ group (Table 1, *p* < 0.01).

## 3. Discussion

hDPSCs have multipotent differentiation characteristics and are therefore regarded as potential sources for the regeneration of dentin under certain circumstances [23]. Odontoblasts secrete several collagenous and noncollagenous proteins, including osteocalcin (OCN), osteopontin (OPN), DSPP), bone sialoprotein (BSP), dentin sialophosphoprotein (DSPP), and dentin matrix acidic phosphoprotein 1 (DMP1), which are regarded as odontogenic differentiation markers of hDPSCs [24,25]. CMET treatment upregulated the expression of these markers, indicating that CMET promoted hDPSCs differentiation towards an odontoblast phenotype. The results from the present study could not confirm the capability of MTA to induce odontogenic differentiation marker expression, which is inconsistent with previous reports [26,27]. The most likely cause for this interesting difference might be the differences in dissolution rate. Previous studies used hard-set disc eluate or transwell inserts; however, to better extract the bioactive content, powder eluate was used in the present study, and the early effects of hard setting materials that have been freshly mixed versus those in their late set stage may differ. In addition, hDPSCs were exposed to MTA for 17 days in the present study, but the MTA action time in the aforementioned reports was 12 to 48 h. Calcium ions might only instantaneously upregulate the mRNA expression of osteo/odontogenic differentiation markers [28], and long-term supplementation with calcium ions resulted in decreased mRNA levels despite increased mineral deposition [29]. Additionally, the effects of aqueous Ca(OH)_2_ and MTA solutions in inducing odontoblastic differentiation declined gradually with time [20]. 

To elucidate the relevant signaling pathways that control the CMET-induced odontogenic differentiation and matrix mineralization of hDPSCs, several possible pathways were assessed. MAPK pathways are well-studied signal transduction pathways activated during odontoblast stimulation in tertiary dentinogenesis [30,31]; thus, targeting MAPK pathways may offer a therapeutic approach in dental disease [32]. In the present study, three specific MAPK inhibitors were chosen: SB202190 is a cell-permeable and highly selective inhibitor of the p38α and β isoforms with a low-half maximal inhibitory concentration that fails to impact other protein kinases; SP600125, a highly selective JNK1/2/3 inhibitor; and PD98059, which blocks extracellular signal-related kinases (ERK) activation by binding MAPK kinase 1 with high specificity, preventing its activation by upstream protein kinases [33,34]. The inhibition of p38 and JNK signaling completely suppressed the matrix mineralization of hDPSCs, which was accelerated by CMET. On the other hand, inhibition of p38 signaling also suppressed CMET-induced ALPase activity to a relatively low level throughout the whole culture period. Inhibition of the JNK pathway reduced the CMET-mediated increase in ALPase activity by one-third on day 14 and to the control level on day 21. However, inhibition of ERK signaling had a less pronounced effect on both the mineralization and ALPase activity of hDPSCs.

Activation of the NF-κB pathway can promote the odontoblastic phenotype and stimulate the odontogenic differentiation of DPSCs [35,36]. BMS-345541 is a highly selective inhibitor of the catalytic subunits of IKK [37]. The IKK complex phosphorylates/degrades IκB and releases NF-κB subunits for nuclear translocation, promoting gene transcription, and regulating diverse biological processes as well as responses to stimuli [38]. In the present study, inhibition of the NF-κB pathway dramatically suppressed the CMET-induced odontogenic differentiation of hDPSCs, as indicated by the downregulation of odontogenic markers and reduced hDPSCs mineral deposition. These results suggest that the NF-κB pathway plays a pivotal role in CMET-induced hDPSCs odontogenic differentiation. 

The aim of the in vivo experiment was to investigate the short-term pulpal response after capping with CMET, especially the effect on hard tissue formation. Although long-term clinical observations of Ca(OH)_2_ treatments are incomparable to tricalcium silicate (TCS)-based materials, their mechanisms are similar. Most TCS-based materials are known to form Ca(OH)_2_ and leach hydroxyl and calcium ions [39]. The release of calcium ions is a pivotal factor for pulp-capping therapies because of the action of calcium on the differentiation and mineralization of hDPSCs [40]. Previous studies have reported that after a few hours of preparation, TCS-based materials release a large amount of calcium ions upon hydration; however, calcium ion release is significantly higher in Ca(OH)_2_ cements after 3 days [41,42]. The solubility of TCS-based cements is reduced by the precipitated calcium phosphate layer. The high solubility and the tunnel defects present in the newly formed dentin bridge turn into major disadvantages of Ca(OH)_2_ and its derivatives, which is less for calcium silicates after a longer setting time [43]. Regarding short-term observation of reparative dentin formation, a previous report suggested that the effects of aqueous suspensions of Ca(OH)_2_ powder and MTA in a rat pulp mechanical exposure model did not significantly differ [44]. The same conclusion was drawn in the comparison of the commercial product Dycal and MTA [45]; therefore, to reduce the use of experimental animals, only Ca(OH)_2_ was used as a positive reference. In the present study, restoration with composite material was preferred over the use of intermediate restorative materials or non-biocompatible materials because adhesive reconstruction provides less toxic and suitable closure. Persistent inflammation delayed the repair process and appears to be an important reason for the lack of complete dentin bridge formation at the exposure site in the negative control group. Although mild to moderate inflammation was observed in the Ca(OH)_2_ and CMET groups, the hard tissue bridges partially prevented the pulp tissues from external stimulation; thus, the original odontoblasts were well preserved. The caustic effect and high pH of Ca(OH)_2_ could be two reasons for the inflammatory response in the Ca(OH)_2_ group; in addition, in all treated groups, unpolymerized resin components may come in contact with pulp tissue after partial dissolution of the capping materials by tissue fluid. These components may be absorbed to some extent by macrophages, allowing antigen passage into the pulp that can maintain at least low-grade inflammation. Thus, the infiltration of dispersed inflammation may occur due to sustained irritation by these components. Whether this inflammation would be resolved over time is unknown. The hard tissue bridge induced by Ca(OH)_2_ was a bone/cementum-like structure. Consistent with the in vitro study, the data collected here also indicated that the effect of Ca(OH)_2_ on calcified tissue formation may be more appropriately regarded as a reparative process than a genuine regeneration response because regeneration refers to the proliferation of cells and tissues to replace lost or damaged cells and tissues with the restoration of normal tissue structure [46]. The findings herein provide evidence that direct capping with CMET can effectively stimulate the odontogenic differentiation potential of DPSCs and promote dentin-like hard tissue formation, and the cells that secrete this structure displayed a polarized odontoblastic characteristic. Additional studies will be necessary to compare the effects of CMET and other materials on cytokine production and the inflammatory response.

The toxicity of monomers is mainly related to their structure, hydrophobicity, and ability to disrupt basic cellular functions. The presence of monomers such as TEGDMA irreversibly exhausts the intracellular glutathione pool, subsequently increasing reactive oxygen species levels and activating major signal transduction pathways that lead to apoptosis in different cell lines [47]. The mechanism of TEGDMA on cell cycle arrest is different from the effect of chemotherapeutic agents [48]. Moreover, the cytotoxicity of TEGDMA might be reduced by N-acetylcysteine (NAC) through the formation of NAC-TEGDMA adduct [49]. A previous study clearly demonstrated that cytotoxic monomers might affect the mitochondrial activity by inducing alterations in energy metabolism, oxidative stress, and glutathione balance even at their subcytotoxic concentrations [50]. Whether the cytotoxic effect of a high concentration of CMET monomer induces the arrest of cell cycle or production of oxidative stress requires future study.

This study had some limitations. First, we used sound teeth and healthy hDPSCs with no signs of inflammation; thus, our data cannot provide information on the effects of CMET on carious teeth with different levels of inflammation. However, it still has the benefit of standardization and can be regarded as acceptable with respect to dentin regeneration. Notably, although the in vivo experiment may have provided valuable information, caution must be exercised when these results are extrapolated to humans. Second, pure CMET monomer powder was used for direct pulp capping, which might not be the ideal format for clinical application; nevertheless, our findings provide evidence that CMET might be a very promising adjunctive for capping materials.

## 4. Materials and Methods

### 4.1. Preparation of Aqueous Solutions of Different Materials

A total of 300 mg of a powder of each material, i.e., CMET (Sun Medical, Gifu, Japan), 4-MET (Sun Medical, Gifu, Japan), Ca(OH)_2_ (Wako, Osaka, Japan), and NEX MTA (GC, Tokyo, Japan) was thoroughly mixed with 30 mL of distilled water (dH_2_O, Invitrogen, Carlsbad, CA, USA) in 50 mL plastic centrifuge tubes using a tube rotator (FINEPCR, Seoul, Korea) at 40 rpm for 72 h at room temperature. The solutions were then centrifuged at 2600× *g* for 15 min, and the supernatants were collected for cell culture. dH_2_O without any solutes served as a control. To prevent bacterial contamination, the aqueous solutions were exposed to ultraviolet irradiation for 60 min before use.

### 4.2. Cell Culture

hDPSCs (PT-5025) were purchased from Lonza. Cells were cultured in Dulbecco’s Modified Eagle’s Medium (DMEM, Sigma-Aldrich, St. Louis, MO, USA) supplemented with 10% fetal bovine serum (FBS, Gibco, Grand Island, NY, USA), 50 units/mL penicillin, and 50 μg/mL streptomycin (Gibco, Grand Island, NY, USA) and inoculated at initial densities of 3 × 10^3^ cells/well in 96-well plates, and 3 × 10^4^ cells/well in 6-well plates. Mineralization-inducing reagents, 10 mM glycerol-2-phosphate disodium salt n-hydrate (β-GP, Wako, Osaka, Japan), 50 μg/mL L-ascorbic acid phosphate magnesium salt n-hydrate (AA, Wako, Osaka, Japan), and the aqueous solutions of each material at 5% (*v*/*v*) were added to the culture medium upon reaching confluence (day 7). The media were changed every 3 days. All cells were cultured in a 37 °C humidified incubator with an atmosphere of 5% CO_2_ and 95% air. Cells at passage 4 were used in this study.

### 4.3. Cell Viability Assay

The cytotoxicity of each material was determined using the Cell Counting Kit-8 assay (CCK-8, Dojindo, Kumamoto, Japan). Materials were directly dissolved in DMEM to obtain high concentrations. hDPSCs were cultured in 96-well plates with DMEM supplemented with 10% FBS, 1% antibiotics, and different concentrations of the materials. CCK-8 assays were performed on day four according to the manufacturer’s instructions. The absorbance of the lysates at a wavelength of 450 nm was measured using a microplate reader (Bio-Rad, Hercules, CA, USA).

### 4.4. ARS Staining

On day 28, cell mineralization was observed by ARS staining (Wako, Osaka, Japan) according to the manufacturer’s instructions. Photographs were taken using a digital imaging system (Funakoshi, Tokyo, Japan) that incorporated an inverted digital camera (Canon, Tokyo, Japan). To quantify calcific staining intensity, the cetylpyridinium chloride (CPC, Sigma-Aldrich, St. Louis, MO, USA) method was used.

### 4.5. ALPase Activity Assay

ALPase activity on days 14, 21, and 28 was measured. Cells were collected and lysed with 0.1% Triton X-100 (Sigma-Aldrich, St. Louis, MO, USA) in dH_2_O, and the lysates were sonicated for 10 min on ice and then centrifuged at 13,200× *g* for 15 min at 4 °C. Then, the supernatants were extracted and subsequently analyzed using an ALPase activity assay (Wako, Osaka, Japan) and a BCA protein assay kit (Thermo Fisher Scientific, Waltham, MA, USA) according to the manufacturers’ instructions, with one unit of enzyme activity defined as the release of 1 nmol of p-nitrophenol per minute at pH 9.8 and 37 °C. The relative activity was determined as follows: units/μg protein = activity (units/μL)/protein concentration (μg/μL). Absorbance at 405 nm and 570 nm was determined using a microplate reader for the ALPase activity assay and protein quantification, respectively.

### 4.6. Real-Time RT-PCR

BSP, OPN, OCN, DSPP, and DMP1 mRNA expression was quantified using primer sets and the real-time RT-PCR conditions provided in Table 2. 

Total RNA was isolated from cultured cells with TRIzol reagent (Invitrogen, Carlsbad, CA, USA) on day 24. The RNA concentration of each sample was measured using a NanoDrop ND-1000 (Thermo Fisher Scientific, Waltham, MA, USA), and one microgram of isolated RNA was then reverse-transcribed into complementary DNA using Moloney murine leukemia virus (Invitrogen, Carlsbad, CA, USA) reverse transcriptase in a 20 μL reaction system according to the manufacturer’s instructions.

Reactions were performed using a LightCycler^®^ Nano (Roche, Basel, Switzerland) according to the manufacturer’s instructions. Target gene expression was normalized to glyceraldehyde 3-phosphate dehydrogenase (GAPDH) gene expression. The comparative 2^−ΔΔCT^ method was used to calculate relative gene expression.

### 4.7. Selective Blockade of NF-κB and MAPK

To investigate the involvement of signal transduction pathways in CMET-treated hDPSCs, inhibitors of p38 isoforms, JNK, and ERK-MAPK pathway (SB202190, SP600125, and PD98059, respectively; 20 μM; Cell Signaling Technology, Danvers, MA, USA) and an NF-κB inhibitor (BMS345541, 1 μM, MedChemExpress, Princeton, NJ, USA) were used. The inhibitors were added to the CMET-containing culture medium when the cells reached confluence.

### 4.8. Direct Pulp Capping and Histological Observation

To evaluate the possibility for clinical CMET application, the maxillary first molars of twenty-four eight-week-old male Wistar rats (Hokudo, Hokkaido, Japan) were used in the in vivo study. The animals were randomly divided into four groups using a random number table: two control groups (blank and negative) and two experimental groups [Ca(OH)_2_ and CMET]; two observation time points were chosen (2 and 4 weeks, *n* = 3). The rats were anesthetized using an intraperitoneal injection of 40 mg/kg pentobarbital sodium (Nacalai Tesque, Kyoto, Japan), and class I cavities on the occlusal surfaces of the maxillary first molars were prepared with #1 and #1/2 round burs (Dentsply, York, PA, USA) with adequate water cooling. To keep the cavities as standardized as possible, all surgical procedures were performed by the same operator with the aid of dental loupes (3.0× magnification, Hogies, VIC, Australia). The depth of the cavity remained constant and was approximately the size of the bur head. The final exposure was performed under pressure using the tip of the endodontic explorer. After irrigation with sterile saline, the exposed site was dried with a sterile cotton pellet until hemostasis was confirmed, and the pulp exposure sites were then directly covered with Ca(OH)_2_ (powder freshly mixed with dH_2_O) or CMET (powder). All cavities were subsequently restored with G-Premio BOND (GC, Tokyo, Japan) and Estelite Universal Flow (Tokuyama Dental, Tokyo, Japan) according to the manufacturers’ instructions. The occlusal surfaces of opposing teeth were adjusted to minimize occlusal forces. All the animals were kept under specific pathogen-free conditions (with food and water) in the animal center. At 2 and 4 weeks after the operation, the animals were euthanized through inhalation of CO_2_ gas, and the whole maxilla was dissected out, fixed in 4% paraformaldehyde for 2 days, and decalcified in a 10% EDTA solution (Hayashi Prue Chemical, Osaka, Japan) with moderate stirring at 4 °C for 10 weeks. The specimens were dehydrated with graded ethanol and embedded in paraffin; then, sagittal serial sections (4 μm thickness) were stained with hematoxylin-eosin to observe histomorphology changes. Two different sections of each specimen (*n* = 6 in each group) were captured and analyzed with the FLOVEL Filing System (FLOVEL, Tokyo, Japan).

Hard tissue formation was determined based on the following subitems, each of which was scored from 1–4, with 1 representing the most desired result and 4 representing the least desired result.

For the continuity of the hard tissue bridge, 1 = complete, 2 = partial/incomplete bridge formation extending to more than one-half of the exposure site but not completely closing the exposure site, 3 = lateral deposition of hard tissue on only the walls of the cavity of pulp exposition, 4 = absence of a hard tissue bridge and absence of lateral deposition of hard tissue.

For the morphology of the hard tissue, 1 = hard tissue bridge with dentinal tubule structure associated with secondary or tertiary dentin, 2 = only irregular hard tissue associated with secondary or tertiary dentin, 3 = only a thin layer of hard tissue deposition, 4 = no hard tissue deposition.

To score the thickness of the hard tissue near the exposure site, the thickest, thinnest, and middlemost point areas of the formed hard tissue were measured. The average of 6 values was calculated: 1 = >200 µm, 2 = 125-200 µm, 3 = <125 µm, and 4 = absent.

The results for these subitems were separately recorded and statistically analyzed, and the data were subjected to the non-parametric tests (Kruskal–Wallis test, followed, if significant, by group comparisons with the Mann–Whitney *U* test). Significance was set at *p* < 0.01.

### 4.9. Statistical Analysis

All in vitro experiments were carried out in triplicate, and the results were expressed as the mean ± standard deviation. Data were subjected to one-way ANOVA and post hoc Tukey’s HSD tests. The statistical significance level was set at *p* < 0.01.

## 5. Conclusions

In summary, this study demonstrates that the novel bioactive monomer CMET has low cytotoxicity and exhibits high activity in stimulating odontogenic differentiation and matrix mineralization both in vitro and in vivo. The findings confirm the underlying signaling pathways involved in this process. Integrating the results of previous in vitro studies, the use of this multifunctional monomer may provide a very promising new therapeutic strategy for material-biology engineering for future endodontic treatments. It is imperative that additional studies be conducted to elucidate the underlying detailed mechanisms, and it would be of interest if longer time intervals were examined to assess the long-term thickness and quality of dentine bridges stimulated by CMET in different species. Additionally, the development of a versatile CMET-containing restorative material used for clinical application is required in future studies.

## Figures and Tables

**Figure 1 ijms-22-12728-f001:**
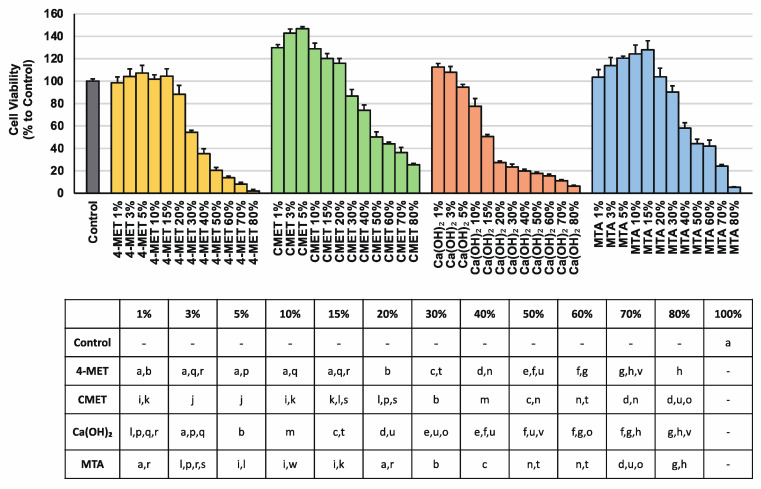
CCK-8 cell viability assay. Different concentrations (*v*/*v*) of aqueous solution of each material were added on day 1, and normal medium without the materials was used as a control. The CCK-8 assay was performed on day 4. Absorbance was measured using a microplate reader; OD = 450 nm. The viable cell ratio of the control group was accepted as 100%, and the other groups were normalized to the control to show differences by percentage compared with the baseline. *n* = 6, different lowercase letters indicate a significant difference (*p* < 0.01).

**Figure 2 ijms-22-12728-f002:**
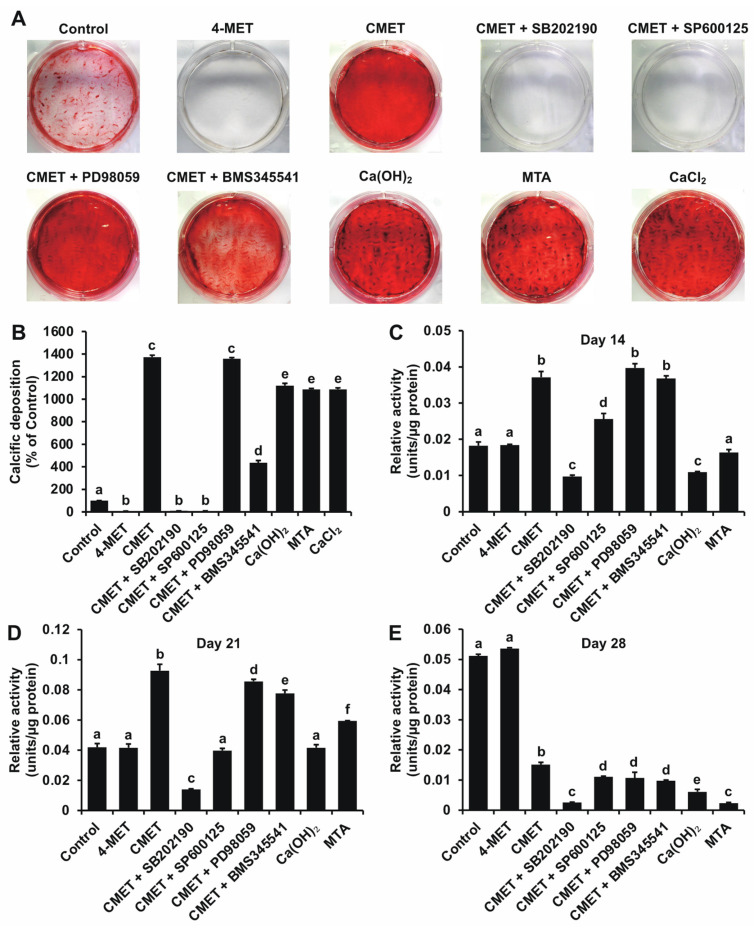
Mineralization-inducing ability of each material. (**A**) Photographs of mineralized nodule formation after 28 days of culture. Except for 4-MET, which inhibited cell mineralization, the other three materials accelerated the calcium nodule formation of hDPSCs. Staining of the CMET group showed a uniform appearance, while staining of the Ca(OH)_2_ and MTA groups was more similar to that upon the addition of a calcium chloride solution. Inhibition of the NF-κB pathway reduced the CMET-induced increase in calcific deposition and inhibition of the p38 and JNK MAPK pathways completely depressed hDPSCs mineralization. (**B**) Mineralized product staining intensity was determined by CPC quantification. The addition of CMET significantly enhanced the ALPase activity of hDPSCs on days 14 (**C**) and 21 (**D**) and then decreased ALPase activity on day 28 (**E**). MTA treatment slightly enhanced hDPSCs ALPase activity on day 21. 4-MET and Ca(OH)_2_ treatment did not increase ALPase activity. CMET-induced ALPase activity was primarily involved p38 and JNK MAPK pathways. *n* = 3, different lowercase letters indicate a significant difference (*p* < 0.01).

**Figure 3 ijms-22-12728-f003:**
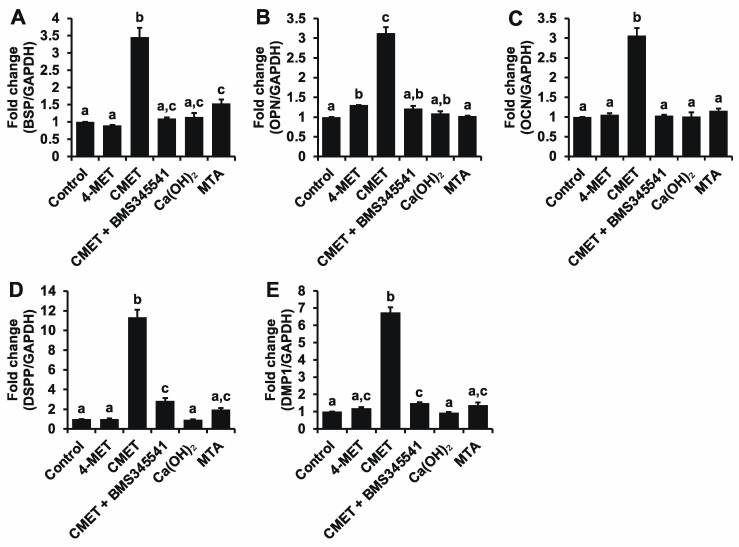
Effects of the aqueous solutions on the mRNA expression of odontogenic differentiation markers. Cells were cultured in DMEM supplemented with 10% FBS and 1% antibiotics. Then, 10 mM β-GP, 50 μg/mL AA, and each aqueous solution at 5% (*v/v*) were added upon confluence, and dH_2_O was employed as a control. The mRNA expression of BSP (**A**), OPN (**B**), OCN (**C**), DSPP (**D**), and DMP1 (**E**) was upregulated by CMET treatment to different extents. These increases were abrogated by the addition of the NF-κB pathway inhibitor BMS345541. GAPDH was used as an internal standard. *n* = 3, different lowercase letters indicate a significant difference (*p* < 0.01).

**Figure 4 ijms-22-12728-f004:**
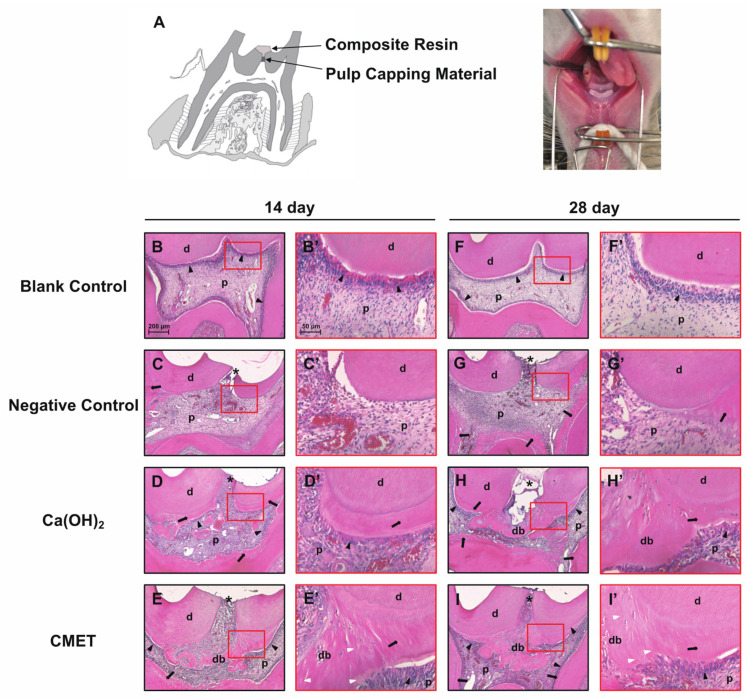
CMET treatment may promote tertiary dentin formation in vivo. (**A**) Schematic of the direct pulp-capping model. Histologic analysis of exposure sites was performed by hematoxylin-eosin staining. The scale bar in the 14-day blank control group image applies to all panels (200 μm and 50 μm in the lower and higher magnification images, respectively). On day 14, severe inflammation and the disappearance of odontoblasts polarization were observed in the negative control group (**C**). Tertiary dentin (arrow) formation and local inflammation near the exposure site were observed in the Ca(OH)_2_ group (**D**). A thin complete hard tissue bridge formed in the CMET group (**E**). Odontoblasts (black arrowhead) remained polarized in the Ca(OH)_2_ and CMET groups. On day 28, heterogeneous reparative dentin was observed in the negative control group, but excluding the exposure site, the pulp tissue remained inflamed (**G**). Continuous hard tissue bridges were found in both the Ca(OH)_2_ and CMET groups (**H**,**I**), but those in the CMET group were homogenous and presented a tubular structure (white arrowhead). (**B′**–**I′**) Higher magnification view of the area demarcated by the open red rectangle in (**B**–**I**). d, dentin; p, dental pulp; db, dentin bridges; * exposure site.

**Table 1 ijms-22-12728-t001:** Results of analysis of hard tissue formation.

	Continuity				Morphology				Thickness			
	1	2	3	4	1	2	3	4	1	2	3	4
**14 days**												
Negative Control	0	0	0	6	0	0	0	6	0	0	0	6
Ca(OH)_2_	0	0	6	0	0	6	0	0	0	0	6	0
CMET	6	0	0	0	6	0	0	0	0	5	1	0
**28 days**												
Negative Control	0	0	0	6	0	0	6	0	0	0	0	6
Ca(OH)_2_	6	0	0	0	0	6	0	0	0	6	0	0
CMET	6	0	0	0	6	0	0	0	5	1	0	0

**Table 2 ijms-22-12728-t002:** Primer sets used for real-time RT-PCR.

Gene	Primer Sequences	Product Size (bp)	Annealing Temperature (°C)
*BSP*	Forward: AAGGGCACCTCGAAGACAAC	119	62.8
	Reverse: CCCTCGTATTCAACGGTGGT		
*OPN*	Forward: TCCCTGTGTTGGTGGAGGAT	158	59.9
	Reverse: GAGTTTTCCTTGGTCGGCGT		
*OCN*	Forward: CGCAGCTCCCAACCACAATA	238	62.8
	Reverse: GTGTGAGGGCTCTCATGGTG		
*DSPP*	Forward: TGCTGGCCTGGATAATTCCG	136	66
	Reverse: CTCCTGGCCCTTGCTGTTAT		
*DMP1*	Forward: ACAGCAGCTCAGCAGAGAGT	235	62.8
	Reverse: TAATAGCCGTCTTGGCAGTC		
*GAPDH*	Forward: CACTAGGCGCTCACTGTTCTCT	250	66
	Reverse: CGTTCTCAGCCTTGACGGT		

## Data Availability

All important data is included in the manuscript.

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
