# Peer review of "Novel Bioactive Adhesive Monomer CMET Promotes Odontogenic Differentiation and Dentin Regeneration"

_ijms, 2021, doi:10.3390/ijms222312728_

Round 1
Reviewer 1 Report
HelloI consider that the subject and the results of this article are
important by their applicability in dental practice.
However, although the edited material is good, I must recommend the authors to
rewrite the article respecting the structure of an article (introduction, purpose,
material and methods ...).
Introduction: the material is well made, maybe a little long.
Aim: it is written clearly and concisely.
Material and methods: I recommend repositioning the written material starting for
example with 4.8. Direct Pulp Capping and Histological Observation (row 349) and
then continue with the other methods of determination used to obtain the results.
Results: the material is well written, the images are clear and well numbered.
Conclusions: Conclusions are found in the abstract but not in the text.
References: there are some references older than 5 years.
Author Response
We are very pleased to note the favorable comments from you. And the authors thank you for your earnest work in reviewing our work. We have made responses according to your comments as listed below:
- Comment: I must recommend the authors to rewrite the article respecting the structure of an article (introduction, purpose, material and methods ...)..
Response: The structure of manuscript has been revised based on the reviewers’ suggestion.
- Comment: I recommend repositioning the written material starting for example with 4.8. Direct Pulp Capping and Histological Observation (row 349) and then continue with the other methods of determination used to obtain the results.
Response: The order of M&M section was followed by conventional pattern of similar publications, and also consistent with the order of aims. Therefore, we decided to move the M&M section after the Introduction section, while remaining the original order of it.
- Comment: Conclusions are found in the abstract but not in the text.
Response: A Conclusions section was added to the revised manuscript.
- Comment: There are some references older than 5 years.
Response: The references which were published more than five years ago are fundamental references for the present study, and our paper is strongly supported by citing these excellent works. We think it is necessary to keep these references in the list for improve the quality of our manuscript.
We hope the changes we have made will meet with your approval.
Reviewer 2 Report
The article is very interesting, the research is nicely performed and the argument could be very interesting for the readers.
However before publication I would suggest some improvements.
1) Move the Materials and Methods section after the Introduction
2) In lines 36-37 you write:
Calcium-containing materials, such as various forms of calcium hydroxide [Ca(OH)2] and mineral trioxide aggregate (MTA), have been extensively used in VPT, and Ca(OH)2 has long been considered the “gold standard”
you could also add a reference explaining that there are several studies that experiment with several experimental materials based on calcium-silicates and calcium-phosphates. One article you could cite is the following one:
Abedi-Amin A, Luzi A, Giovarruscio M, Paolone G, Darvizeh A, Agulló VV, Sauro S. Innovative root-end filling materials based on calcium-silicates and calcium-phosphates. J Mater Sci Mater Med. 2017 Feb;28(2):31. doi: 10.1007/s10856-017-5847-1. Epub 2017 Jan 20. PMID: 28108959.
3) Inlines 55-58 you write:
A previous study revealed that most of monomers used in the adhesive system and composite resin, such as 2,2-bis[4-(3-methacryloxy-2-hydroxy- propoxy) phenyl]propane (bis-GMA), triethylene glycol dimethacrylate (TEGDMA) and 2-hydroxyethyl methacrylate (HEMA), showed cytotoxic effects on odontoblast-like cells in vitro [14].
The authors could outline more that cytotoxic effects can depend not only on the materials but also on the conversion within the environment. You could cite the following reference to support this sentence:
Marigo L, Nocca G, Fiorenzano G, Callà C, Castagnola R, Cordaro M, Paolone G, Sauro S. Influences of Different Air-Inhibition Coatings on Monomer Release, Microhardness, and Color Stability of Two Composite Materials. Biomed Res Int. 2019 May 9;2019:4240264. doi: 10.1155/2019/4240264. PMID: 31211136; PMCID: PMC6532316.
4) Include a Conclusion section
5) Describe limitations of your study and future research needed.
Author Response
The authors thank the reviewer for the constructive comments and thoughtful suggestions, and we appreciate the time and effort you have put into your comments. We have made corrections according to your suggestions as listed below:
- Comment: Move the Materials and Methods section after the Introduction.
Response: In the revised manuscript, the M&M section was moved after the Introduction section.
- Comment: In lines 36-37 you write: Calcium-containing materials, such as various forms of calcium hydroxide [Ca(OH)2] and mineral trioxide aggregate (MTA), have been extensively used in VPT, and Ca(OH)2 has long been considered the “gold standard”.
You could also add a reference explaining that there are several studies that experiment with several experimental materials based on calcium-silicates and calcium-phosphates. One article you could cite is the following one: Abedi-Amin A, Luzi A, Giovarruscio M, Paolone G, Darvizeh A, Agulló VV, Sauro S. Innovative root-end filling materials based on calcium-silicates and calcium-phosphates. J Mater Sci Mater Med. 2017 Feb;28(2):31. doi: 10.1007/s10856-017-5847-1. Epub 2017 Jan 20. PMID: 28108959.
Response: Based on the suggestion, we have added this citation in the revision.
- Comment: Inlines 55-58 you write: A previous study revealed that most of monomers used in the adhesive system and composite resin, such as 2,2-bis[4-(3-methacryloxy-2-hydroxy- propoxy) phenyl]propane (bis-GMA), triethylene glycol dimethacrylate (TEGDMA) and 2-hydroxyethyl methacrylate (HEMA), showed cytotoxic effects on odontoblast-like cells in vitro [14].
The authors could outline more that cytotoxic effects can depend not only on the materials but also on the conversion within the environment. You could cite the following reference to support this sentence: Marigo L, Nocca G, Fiorenzano G, Callà C, Castagnola R, Cordaro M, Paolone G, Sauro S. Influences of Different Air-Inhibition Coatings on Monomer Release, Microhardness, and Color Stability of Two Composite Materials. Biomed Res Int. 2019 May 9;2019:4240264. doi: 10.1155/2019/4240264. PMID: 31211136; PMCID: PMC6532316.
Response: Based on the suggestion, we have added this citation in the revision.
- Comment: Include a Conclusion section.
Response: A Conclusions section was added in the revised manuscript.
- Comment: Describe limitations of your study and future research needed.
Response: The limitations of the present study was located at the end of discussion section, and further research was described in the Conclusions section.
Thank you for all your detailed comments and suggestions. We have endeavored to make careful modifications on the original manuscript based on your comments. We hope the revision will meet with your approval.